# Non-Coding RNAs as Regulators and Markers for Targeting of Breast Cancer and Cancer Stem Cells

**DOI:** 10.3390/cancers12020351

**Published:** 2020-02-04

**Authors:** Kirti S. Prabhu, Afsheen Raza, Thasni Karedath, Syed Shadab Raza, Hamna Fathima, Eiman I. Ahmed, Shilpa Kuttikrishnan, Lubna Therachiyil, Michal Kulinski, Said Dermime, Kulsoom Junejo, Martin Steinhoff, Shahab Uddin

**Affiliations:** 1Translational Research Institute, Academic Health System, Hamad Medical Corporation, Doha 3050, Qatar; hamnafathima2001@gmail.com (H.F.); emoibrahim04@gmail.com (E.I.A.); SKuttikrishnan@hamad.qa (S.K.); LTherachiyil@hamad.qa (L.T.); MKulinski@hamad.qa (M.K.); MSteinhoff@hamad.qa (M.S.); skhan34@hamad.qa (S.U.); 2National Center for Cancer Care and Research, Hamad Medical Corporation, Doha 3050, Qatar; ARaza@hamad.qa (A.R.); SDermime@hamad.qa (S.D.); 3Sidra Medicine, Doha 26999, Qatar; tkaredathabdulazis@sidra.org; 4Department of Stem Cell Biology and Regenerative Medicine, Era University, Lucknow 226003, India; drshadab@erauniversity.in; 5Qatar College of Pharmacy, Qatar University, Doha 3050, Qatar; 6General Surgery Department, Hamad General Hospital, Hamad Medical Corporation, Doha 3050, Qatar; KJunejo@hamad.qa; 7Department of Dermatology Venereology, Hamad Medical Corporation, Doha 3050, Qatar; 8Department of Dermatology, Weill Cornell Medicine, Qatar Foundation, Education City, Doha 24144, Qatar; 9Department of Medicine, Weill Cornell Medicine, New York, NY 10065, USA

**Keywords:** breast cancer stem cells, biogenesis, long non-coding RNA, microRNA, targets

## Abstract

Breast cancer is regarded as a heterogeneous and complicated disease that remains the prime focus in the domain of public health concern. Next-generation sequencing technologies provided a new perspective dimension to non-coding RNAs, which were initially considered to be transcriptional noise or a product generated from erroneous transcription. Even though understanding of biological and molecular functions of noncoding RNA remains enigmatic, researchers have established the pivotal role of these RNAs in governing a plethora of biological phenomena that includes cancer-associated cellular processes such as proliferation, invasion, migration, apoptosis, and stemness. In addition to this, the transmission of microRNAs and long non-coding RNAs was identified as a source of communication to breast cancer cells either locally or systemically. The present review provides in-depth information with an aim at discovering the fundamental potential of non-coding RNAs, by providing knowledge of biogenesis and functional roles of micro RNA and long non-coding RNAs in breast cancer and breast cancer stem cells, as either oncogenic drivers or tumor suppressors. Furthermore, non-coding RNAs and their potential role as diagnostic and therapeutic moieties have also been summarized.

## 1. Introduction

Breast cancer (BC) is the most common form of cancer among women and accounts for 11.6% of cancer incidences and 6.6% of cancer-associated deaths worldwide [1]. The high incidence and death rates in BC are linked to various factors, among which the most common being its heterogeneous nature. The inter/intratumoral heterogeneity, usually affecting one anatomic site of the breast with phenotypic and molecular diversity, plays a key role in its histology and staging [2,3]. Previously, histological stratification of BC was based primarily on the expression status of hormonal receptors, such as the estrogen receptor (ER), progesterone receptor (PR), and ERBB2 receptor (HER2) [4]. However, with advances in molecular analysis and gene expression profiling, further subtypes of BC, including luminal ER positive (luminal A and luminal B), HER2 enriched and triple negative (basal like) have been identified [5]. This molecular sub-classification has served as a guiding principle for the utility of targeted therapies such as synthetic lethality using poly ADP ribose polymerase (PARP) inhibitors HER2-targeted (e.g., Trastuzumab) and hormonal (e.g., Tamoxifen) therapies, leading to better outcomes and management of BC [5]. Several organizations including the American Society of Clinical Oncology (ASCO) and National Comprehensive Cancer Network (NCCN) have also issued extensive recommendations and guidelines for implementation of molecular analysis as a tool for risk stratification, treatment planning and management [6,7,8]. 

Currently, the individualized treatment strategy is based on various factors including tumor size, morphology, grade, metastases, ER, PR and HER2 expression [9]. While detailed information about these factors is critical for therapeutic management, identification and understanding of these diagnostic/predictive markers will aid in implementing personalized treatment strategies. Therefore, breakthrough data on transcriptional regulators of gene expression, known as “non-coding RNA” has become a focus of research worldwide.

The transcriptome of most organisms is far more complex than originally imagined, as the vast majority of genomic sequence is extensively transcribed into a diverse range of protein coding and non-coding RNAs (ncRNAs) [10]. Surprisingly, out of 75% of the transcribed human genome, only about 2% represents the protein coding region [11]. Until recently, the majority of the transcriptome which lacks coding potential was considered to be “Junk” or products of faulty aberrant splice events [11]. Considerable improvements in high-throughput technologies, such as RNA sequencing, have allowed the identification of several previously unannotated non-protein coding transcription events in genomic regions. The efforts for re-evaluating non-coding part of the human genome and re-classifying them from “junk” to “non-junk” have been accomplished mainly through the Encyclopedia of DNA Elements project (ENCODE) project and by using ab initio transcriptome assembly which provides unbiased modality for lncRNA discovery which can pinpoint cancer- associated ncRNAs [12,13]. These projects provided critical insights into the “junk” or “dark matter” of DNA being transcribed via complex regulatory networks for the regulation of coding genes. Thus, the pinnacle of interest was shifted from coding genes to transcripts as the fundamental units of the genome. 

The classification of the non-coding part of the genome, known as ncRNAs, is based on their length. Keeping the cutoff at 200 nucleotides’ length, the ncRNAs <200 nucleotides are designated as short noncoding RNAs (sncRNAs). These include microRNA (miRNA), small interfering Ribonucleic Acid (siRNA), piwi-interacting RNA (piRNA), small nucleolar RNAs (snoRNAs), small nuclear RNA (snRNA), and tRNA-derived fragments (tRFs) [14]. The ncRNAs >200 nucleotides, known as lncRNAs [15] include intronic, antisense, long intervening/intergenic noncoding RNAs (lincRNA), competing endogenous RNA (ceRNA), etc. [16]. Both miRNAs and lncRNAs can control fundamental cellular and biological processes via diverse mechanisms and have been associated with playing key regulating roles in transcriptome by establishing networks and interactions. Since miRNAs are considered to be negative regulators of gene expression, lncRNAs are also considered to be an important regulator in different ways of gene expression including cross-talk with miRNA, sponging the microRNA, and regulating their expression [17,18,19]. The expression and function of miRNAs and lncRNAs are tightly regulated and conserved in development and physiological homeostasis. The role of miRNAs and lncRNAs is critical and leads to the pathogenesis of various human diseases such as cancer by dysregulation of human transcriptome [20]. 

The miRNAs are small, 18–23 nucleotide long transcripts involved in gene regulation via post-translational modifications [21]. The mechanism of action of miRNA involves interacting by binding to the open reading frame or to the 3’ untranslated regions (3′ UTRs) of target genes, which leads to repression of gene expression of the translating mRNA or mRNA degradation through formation of functional complexes via activation of Argonaute (Ago) proteins which target the 3′ UTRs [22]. The biogenesis of miRNAs is shown in detail in Figure 1. Numerous studies documented the role of miRNA in cancer progression. Oncogenic miRNAs are associated with regulation of tumor suppressor genes and targeting of oncogenes thus promoting invasion, metastasis, and drug resistance [23]. 

In addition to miRNAs, lncRNAs [24,25] were been reported for their functionally important roles in cancers [16,26]. The biogenesis of lncRNA is a complex process with capping, splicing, and polyadenylation [27,28]. The main mechanisms include cleavage by ribonuclease P (RNaseP) to generate 3′ mature ends [29], the formation of snoRNA and snoRNP complex caps at the ends, and finally special 5′- and 3′ end processing to convert it into a circular stable structure [30,31,32] (Figure 2). Recently, unique sub-nuclear structures, known as “paraspeckles”, with protein-rich nuclear organelles around a specific lncRNA scaffold, were identified during biogenesis [33]. They have been said to stimulate gene regulation through sequestration of component proteins and RNAs, with subsequent depletion in other compartments [34]. 

The ENCODE project identified more than 28,000 unique lncRNAs, most of which are still not properly annotated or identified [35]. Functional characterization of several of them is still a challenge except in the case of some classically defined important lncRNAs which are well explored, such as X inactive specific transcript (XIST; in X chromosome inactivation), oncogenic lncRNA HOX Transcript Antisense Intergenic RNA (HOTAIR); in positional identity and telomerase RNA component (TERC; in telomere elongation), ANRIL a lncRNA in molecular scaffold of chromatin-modifying complexes, decoy RNAs such as GAS5 (growth arrest specific 5) and TERRA (telomeric repeat-containing RNA) [36,37]. A plethora of regulatory functions were unveiled in several lncRNAs which affects their cellular functions associated with development and pathophysiology of diseases including several types of cancer, neurological and cardiovascular conditions, and immunological and metabolic disorders [38,39,40]. 

Published data underpinned the roles played by miRNA and lncRNA in invasion and metastasis in BC and Breast cancer stem cells (BCSCs). However, a detailed study on the interaction of ncRNA with cancer stem cells (CSCs) and their effects on metastasis and recurrence has not yet been carried out. Our present review aims to outline research studies that highlight the impact of miRNAs and lncRNAs on tumor occurrence and progression in BC and BCSCs, while also underscoring the potential role governed by ncRNAs as diagnostic and therapeutic moiety that may lay as future foundation in development of newer strategies to prevent and overcome issues related to invasion and metastasis in BC and BCSCs. 

## 2. BCSCs and Their Regulation 

CSC is a small population that exhibits characteristics of both cancer cells and stem cells including self-renewal, differentiation, asymmetric/symmetric division, as well as alterations in their gene expression. CSCs have the ability to seed tumors when transplanted into an animal host as well as give rise to non-CSC bulk tumors in order to promote disease progression [41,42]. Therefore, BCSCs represent a heterogeneous population of cancer cells that possess the ability to form transplantable tumors, tumor maintenance, progression, therapeutic resistance, and relapse [43]. Characterization of BCSC has shown that they express a panel of markers depending on their source of derivation. For example, when isolated from transgenic mouse models, BCSC tend to express CD133^+^, CD24^+^ Thy1^+^, CD29^lo^ CD24^+^ CD61^+^, Sca1^+^, CD24^+^ CD29^+^/CD49f^+^ whereas when isolated from cell lines, the main markers for identification include MUC1^+^, Procr^+^/ESA^+^, CD49f^+^/DLL1^hi^/DNER^hi^, GD2^+^, CD44^+^/CD24^−^/^lo^/ANTXR1^+^, ABCG2^+^, Lgr5^hi^, CD44^+^CD24^-^/^lo^SSEA-3^+^ or ESA^hi^PROCR^hi^SSEA-3^+^, Nectin-4^+^ and CD70^+^ [44]. However, the most widely used markers for identification are CD44/CD24 and ALDH1 [45]. It has been reported that tumors expressing even a small number of cells with CD24^−^/CD44^+^ and ALDH1^+^ markers exhibit an increased tumor-initiating capacity in NOD/SCID mice [46] indicating the significance of these two distinct subtypes in BCSC characterization especially with respect to their location and proliferation capability [45]. In BC, mesenchymal-epithelial transition (MET) CSCs bears higher ALDH expression as well as higher proliferation rate is contrary to epithelial-mesenchymal transition (EMT) CSCs which are enriched with CD44^high^/CD24^-^ expression but with poor proliferation rate. However, aggressive clinical behavior in tumor types is attributed to the high proportion of ALDH-expressing CSCs [45,47]. 

## 3. BCSCs and Tumor Microenvironment

The normal breast tissue is highly heterogeneous and has the unique capacity to self-renew/regenerate, proliferate and differentiate into mature luminal and myoepithelial cells with the help of mammary stem cells (MaSCs) that reside within the microenvironment [48,49]. The regulation of these MaSCs is dependent upon the components of the microenvironment including blood vessels, immune cells, signaling molecules, fibroblasts, and the extracellular matrix (ECM) [48,50,51]. Similarly, in BC, the tumor microenvironment (TME), consisting of cancer-associated fibroblasts (CAFs), MSCs, immune cells, immune-suppressive cells, endothelial cells, cytokines, growth factors, etc. are known to play a critical role in the regulation and modulation of BCSCs thus facilitating therapeutic resistance, metastasis, and progression [52]. 

The role of various components of the TME in BCSCs activity is documented in several studies [53,54,55]. For example, CAFs within the microenvironment release several growth factors, hormones like platelet-derived growth factor-BB, cytokines, and chemokines, such as CCL2, CCL7, IL-6 and IL-8, that modulate CAFs and promote stemness and expansion of BCSC [55,56,57,58,59]. CAFs are considered to be a central core component in the maintenance of CSC properties thereby promoting stemness in BC cells [60,61,62,63]. Similar to CAFs, another important component of the tumor stroma that plays a role in the expansion of BCSCs is MSCs [53]. Studies reported that MSCs regulate increased production of CXCL7 and IL-6 via positive feedback mechanism that promotes BCSC self-renewal, expansion as well as metastatic potential [64].

In addition to CAFs and MSCs, a variety of immune cells including T cells, macrophages, and T regulatory cells (Tregs) also play a critical role in the modulation of TME to promote the expansion of BCSCs [65]. In the past, several studies have reported that tumor-associated macrophages (TAMs) are commonly involved in the expansion of BCSCs via the up-regulation of HAS2 (hyaluronan synthase) and paracrine EGFR/STAT3/SOX-2 signaling pathway [66,67]. In addition to this, TAMs promote the secretion of cytokines including IL-6, IL-8, GM-CSF, TNF-α and TGF-β that allows regulation, maintenance, and proliferation of BCSCs [52,68]. 

## 4. Regulatory Pathways Associated with BCSC

The regulation of BCSCs is largely dependent on key signaling pathways including JAK/STAT, Notch, Wnt, and Hedgehog [69,70,71,72]. The dysregulation of these pathways facilitate differentiation and self-renewal of BCSCs leading to increased proliferation, invasion, and metastasis in cancers [69,73].

Accumulating evidence suggests that dysregulation of the JAK/STAT3 pathway is the common mechanism involved in the maintenance/regulation of BCSCs [74,75]. In BC, the modulation of TME via secretion of cytokines, growth/transcription factors including IL6/STAT3, NO/NOTCH, Twist2 and hormones such as leptin facilitate activation/phosphorylation of JAK/STAT3 pathways leading to enhanced self-renewal and differentiation capacity in BCSCs [76,77,78]. In addition to this, studies have reported that the activation of JAK/STAT3-Regulated Fatty Acid β-Oxidation I (STAT3-CPTIB-FAO) and EGFR/STAT3/SOX-2 paracrine signaling also play an important role in conferring drug resistance -associated characteristics to BCSCs thus leading to treatment failures [66,79]. Another signaling pathway that is known to be involved in the maintenance and self-renewal of BCSCs is the Notch signaling pathway [69,70]. This pathway is activated via binding of Notch receptors to Notch ligands thus leading to translocation of the Notch intracellular domain (NCID) to the nucleus. The subsequent hyperactivation of downstream effector molecules regulates the asymmetric division and self-renewal of BCSCs [69,70]. Increased levels of Notch1 are associated with increased ALDH1 levels in BCSCs indicating that Notch signaling dysregulation is important for BCSC proliferation and maintenance [80]. Reports also suggest that the expansion of BCSCs is influenced by several factors such as histone-lysine N-methyltransferase (Enhancer of Zeste Homolog 2; EZH2) and lipid mediator sphingosine-1-phosphate (S1P). Increased levels of EZH2 and SIP enhance NOTCH1 activation and signaling leads to increased tumorigenic ability in mice and breast cancer patient- derived mammospheres [81,82].

The Wnt/Frizzled/-catenin signaling is a critical pathway that activates Wnt-targeted transcription factors via nuclear translocation of cytosolic b-catenin. This, in turn, facilitates activation of Wnt-targeted genes through binding to the T cell factor/lymphoid enhancing factor family (TCF/LEF) leading to activation of genes associated with cellular differentiation, asymmetric division and cell migration [74,83]. In BCSCs, activation of Wnt signaling due to transcription factor Sry-related HMG box 9 (Sox9) supported stemness and increased mammosphere-formation in BC cell lines thus suggesting that increased Wnt signaling is associated with enhanced BCSC proliferation, self-renewal, and maintenance [84]. 

The Hedgehog pathway is also an important signaling pathway that is activated via smoothened that facilitates cytoplasmic translocation of Gli-com to the nucleus [69]. Like Wnt signaling pathway aberrant activation of Hedgehog pathway due to overexpression of smoothened or due to various growth factors (fibroblast growth factor 5 (FGF5) and collagen), EMT, MET, CAF have been observed to be involved in maintenance, proliferation and, stemness of BCSCs [60,61,65,85,86,87] Therefore, the Hedgehog pathway is considered to be an important regulatory pathway for maintenance of stemness in breast cancer cells [69]. 

## 5. Role of MicroRNAs and LncRNA in BCSCs

MicroRNAs, including oncomiRs and Tsmirs, have been critically implicated in the regulation of BC development and progression via regulatory networks. Modulation of signaling pathways such as PI3 kinases, Wnt/βcatenin, STAT, HIF 1α, etc. by miRNAs directly or indirectly influences hallmarks of cancers and facilitates tumor suppression/progression [88]. Studies have shown that functional interaction of miRNA with cell proliferation and cell cycle progression factors such as cyclin protein families, protein kinases, etc. serves as an important target for tumor suppression/proliferation in BC [88]. For example, miRNAs, such as miR-497, miR-16, and miR-30c-2-3p, were reported to target and inhibit cell cycle regulator of G1-S transition, cyclin E1 leading to decreased cyclin E1 expression and suppression of proliferation by blocking BC cells from entering the S-phase of the cell cycle [89,90,91,92]. On the other hand, certain miRNAs, such as miR-483-3p, dysregulate the cell cycle transition by facilitating the formation of cyclin E1 and cyclin-dependent kinase CDK2 complex. This leads to increased expression of cyclins, up-regulation of protein kinases and down-regulation of kinase inhibitors, thereby increasing BC cell viability and proliferation [92]. Similarly, overexpression of miR-1207-5p, has been associated with negative regulation of STAT2 expression and inactivation of cell cycle-dependent kinase inhibitors CDKN1A and CDKN1B thus promoting cell cycle progression in cancer cells [93].

The WNT/β catenin pathway is a well-documented target of miRNAs. Various studies have shown that modulation of this pathway can affect the migration/invasive potential of BC cells [88]. For example, overexpression of miR-148a has been reported to decrease migration of BC cells via targeting of WNT-1 ligand of the WNT/β catenin pathway. This leads to reduced levels of WNT-1 mRNA/protein, catenin, MMP-7, and TCF-4 levels, thus affecting the migration of cancer cells [94,95]). miR-340, has also been identified as a regulator of the WNT/β catenin pathway and acts to influence migration/invasion of BC cells via molecular targeting of associated genes such as c-MYC, CTNNB1and ROCK1 [95]. Furthermore, other signaling molecules, suppressed by miRNAs, include SMAD7, MTA1, WT1, SETBP1, EphA4, LASP1, and STAT3. Suppression of these molecules via down-regulation of miRNAs including miR-497, miR-421, miR-193a etc. leads to reduced migration/invasion potential of BC cells [96,97,98].

In addition to the regulation of the Wnt/β catenin pathway, certain miRs have also been identified to regulate the PI3K/Akt signaling pathway [99,100]. For example, miR-204-5p is important in BC as its overexpression leads to a reduction in cell proliferation, migration, and metastasis via direct inhibition of PIK3CB. Furthermore, it is also involved in modulation of key immune cells such as myeloid-derived suppressor cells (MDSCs), macrophages, and natural killer (NK) cells to supports cancer cell proliferation via remodeling of tumor microenvironment [101]. 

Like in BC cells, miRNAs are associated with directing their oncogenic/suppressor potential in BCSCs (Figure 3, Table 1) [102]. For example., miR-200 family comprising of miR-200a, miR-200b and miR-200c [103] is well-known for their extensive role in conferring stem cell-like properties in BC cells including mammospheres formation, EMT regulation, metastasis, invasion, apoptosis, survival, and cancer cell growth [103,104]. There are various mechanisms by which miR-200b and miR-200c modulate target genes in order to facilitate stem cell-like properties. For instance stem cell transcription factor KLF4, suppressor of zeste 12 (SUZ12), poly-comb complex protein BMI1 and Prolyl isomerase Pin1 are frequently targeted by miR-200c leading to transcription repression and influencing BCSC formation [105,106,107]. On the other hand, the up-regulation of miR-200 decreases the expression of ZEB1/ZEB2 leading to reduced expression of E-cadherin and affecting the metastatic potential of BCSCs [108,109]. Similarly, studies documented that increased expression of miR-200c via direct binding of tumor suppressor tumor protein p53 (p53) leads to decreased stem cell properties in BC [110]. Furthermore, knockdown of miR-200 was reported to promote mammosphere-formation via direct targeting of the ten-eleven translocation (TET) family and leading to enhanced metastasis in a mouse xenograft model [111]. In addition to this, EGF-driven invasion was also reported to be regulated and controlled by the miR-200 family [104]. 

Another miRNA family that plays an important role in BCSCs is the miR-34 family. Studies have shown that miR-34 family members, usually activated by p53 [182], are well-known to influence CSC such as properties in BC [165,183,184]. Their mechanism of action is via meditation of cell cycle arrest/apoptosis [182] as well as targeting of various signaling pathways such as BCL-2, CCND1 MYC, E2F3 CDK6, SIRT1, and Notch1/4 leading to negative regulation of cell proliferation, invasion, migration, and subsequent inhibition of BCSCs propagation [183,185,186,187]. Similarly, a study on BC patient tissues has shown that miR-34 is negatively correlated with tumor stages and metastasis indicating its role in breast cancer progression [188]. Furthermore, overexpression of miR-34a and miR-34c has been documented to reduce mammospheres formation, inhibit the development of CD44^+^CD24^-^/ALDH^+^ cells as well as eradicate BCSCs [183,184,188].

Guarnieri et al., has reported on a novel mechanism of the miR-106b-25 cluster as a regulator of breast tumor initiation and BCSC phenotypes [189]. The results of the study show that overexpression of miR-106b-25 cluster targets repression of NEDD4L thus leading to increased NOTCH1 signaling and enhanced stem cell phenotypes in tumor imitating cells both vitro and in vivo. These results were further validated in metastatic breast cancer patient samples [189]. Similarly, the overexpression of the miR-125 family has also been associated with the modulation of stem cell-like properties in BC via targeting of receptor tyrosine-protein kinase 2/3 and Eukaryotic Translation Initiation Factor 4E Binding Protein 1 (ErbB2/3and EIF4EBP1) [190]. Overexpression of miR-125 enhances BC progression by increasing the expression of oncogenes. Therefore, miR-125 families are considered to be potential therapeutic targets [103]. Overexpression of miR-181family members via different molecular mechanisms have been associated with facilitating BCSCs in mammospheres formation, self-renewal, colony formation, tumor development as well as with poor prognosis in TNBC patients [117,189,190,191,192]. Additionally, inhibition of miR-181a/b via targeting of the Pleckstrin homology-like domain, family A, member1 (PHLDA1) has demonstrated a reduction in mammospheres formation in BC cells [193]. Furthermore, miR-27 is reported to be an important regulator of BCSCs and functions via targeting various immune mechanisms. The main mechanisms influenced by miR-27 are regulation of macrophages, activation of NF-kappaB /MAPK pathways and reduced dendritic cell-mediated differentiation of Th1 and Th17 cells [194,195]. This was shown in BC patients wherein a decrease in the miR-23a/27a/24-2 cluster in TAMs enhanced tumor growth and vice versa [196]. In addition to this, RUNX1 mediated transcriptional up-regulation of miR-27a is associated with differentiation of BCSC into endothelial cells and targeting of signaling pathways ZBTB10, MYT-1. This was reported to play a significant role in modulation of proliferation, self-renewal ability, angiogenesis, metastasis and enhanced tumorigenicity in BC cells [197].

There are a vast number of miRNAs that have been reported to be involved in the regulation of BCSCs via targeting various pathways. In addition to the ones discussed above, some of the important ones also include miR888, miR-30 family, miR-16, Let-7 family, miR-140-5p, miR-205, miR-495, etc. Overexpression or inhibition of such miRNAs can regulate the expansion of BCSCs, conversion from non-stem to stem cell phenotype, self-renewal, promotion of colony formation and affecting the number and size of mammospheres [165].

The human genome comprises 17,910 lncRNA that are often overexpressed or down-regulated in BC at various levels [198,199]. Some of the lncRNAs found to be associated with initiation, progression, and metastasis in BC include HOTAIR, Small nucleolar RNA host gene 12 (SNHG12), Long intergenic non-coding RNA for kinase activation (LINK-A), Rhabdomyosarcoma 2-associated transcript (RMST), RMRP (RNA component of mitochondrial RNA processing endoribonuclease), nuclear paraspeckle assembly transcript 1 (NEAT1), steroid receptor RNA activator (SnaR), MALAT1 (metastasis-associated lung adenocarcinoma transcript 1), CCAT2 (Colon Cancer Associated Transcript 2), CRNDE (colorectal neoplasia differentially expressed), MIAT (myocardial infarction associated transcript), MEG3 (Maternally Expressed 3), CAT104, LINC01234, STXBP5-AS1, RMRP, GATA3-AS1, RP11-279F6, AC017048 and LINC-ROR. [199,200,201,202].

In CSCs, several lncRNA such as ROR, HOTAIR, H19, UCA1, and ARSR were reported to play a significant role in stemness, proliferation, invasion, and migration via targeting of signaling pathways/sponging of various microRNA through competing for endogenous RNA (ceRNA) [199,203]. For e.g., lncRNA CRNDE was reported to be up-regulated via sponging and subsequent repression of miR-136 expression in BC cell line, MDA-MB231 as well as in BC tissues [204]. The study observed that CRNDE overexpression was associated with activation of Wnt/β-catenin, c-myc and cyclinD1 signaling pathways thus facilitating stemness, cell proliferation, migration, and invasion. Similarly, overexpression of CRNDE in mouse models showed an increase in tumor weight and volume indicating its role in promoting tumorigenesis [204]. 

lncRNA HOTAIR is a well-studied lncRNA and is reported to manifest carcinogenic potential such as migration, metastasis, invasion, EMT transition, and stemness in cancerous cells mainly via regulation of gene silencing [201]. Mir-7 by targeting the SETDB domain inhibited cellular processes, decreased the population of BCSCs and also partially reversed EMT through suppression of the STAT3 pathway in MCF-7, MDA-MB-231 cell lines and in BCSC xenograft model [205]. Furthermore, a study on CSCs of MCF-7 and MDA-MB-231 reported that HOTAIR influences migration, self-renewal, and colony formation in BCSCs via transcriptional inhibition of miR-34a and subsequent up-regulation of SOX 2. The authors validated the association of HOTAIR on functional regulation of miR-34a in BCSCs by introducing miR-34a mimics plus HOTAIR in CSCs. The results showed reduced proliferation potential of HOTAIR, thus evidencing the link between miR-34a and HOTAIR in BCSCs self-renewal and proliferative ability. On the other hand, modulation of full length HOTAIR expression was found to be associated with negative regulation of miR-34a indicating that full length HOTAIR is essentially required to affect miR-34a regulation, self-renewal, and colony formation capacity in BCSCs. In addition, up-regulated HOTAIR was also found to be involved in p53 induction thus affecting proliferation and colony formation in CSCs [206].

Another lncRNA known as lncRNAH19 is reported to be essentially involved in the induction of BC cell stemness, migration and mammosphere-formation. It functions mainly by acting as a ceRNA for miR-let 7 with subsequent overexpression of LIN28, HIF 1α, and PDK1. Since these markers are involved in inducing stem-like phenotypes, their role in BCSCs is deemed critically important. Studies on BC tissues and samples have also reported on high levels of lncRNAH19 and investigation on knockdown of H19 in nude mice has evidenced suppression of tumor growth indicating the significance of lncRNAH19 in BC tumorigenesis [207,208]. Similarly, LINC00511, a ceRNA for mir185-3p, has also been associated with influencing stemness in BCSCs. It functions by targeting E2F1 protein which in turn binds to Nanog promotor, thus forming a LINC00511/miR-185-3p/E2F1/Nanog axis leading to maintenance of BCSCs, enhanced mammosphere-formation and promotion of cell proliferation and invasion [209]. 

The TME plays an influential role in the induction of stem cell-like properties in BC cells through lncRNA. In TNBC, MSC and CAF trigger up-regulation of LINC01133 thereby inducing signaling of pluripotency factor Kruppel-Like Factor 4 (KLF4) and promoting CSC like phenotypic properties in BC cells [210]. LINC00284, another important lncRNA in TNBC has recently been identified as non-coding RNA in the aldehyde dehydrogenase 1A pathway (NRAD1) and has been documented to be functionally associated with CSCs in TNBC. This functional association and significance are based on two observations; firstly, it has been found to have genomic interactions (in the intronic regions) and secondly it is directly regulated by CSC marker ALDH1A3. This strong association indicates that NRAD1 is an important mediator of breast cancer cell proliferation and survival [211].

LncRNA RoR (regulator of reprogramming) is considered to be an important regulator of pluripotent stem cells via targeting of transcription factors SOX2, OCT4, NANOG and sponging of miR-145 [212,213,214]. As a ceRNA of mir-145, ROR functions via loss of mature miR-141 expression leading to the protection of pluripotency factors [213]. In BC cells and in patient samples, lncRNA ROR was not only linked to the self-renewal of stem cells, EMT transition, and drug resistance but also to poor prognosis indicating its significance in tumorigenesis process [215,216,217,218,219,220]. Mainly, lnc-ROR functions via targeting of ZEB1/2 and TGF-β signaling leading to modulation of EMT markers such as vimentin and neural (N)-cadherin and induction of EMT process [218,219,221,222]. Furthermore, studies on silencing/knockdown of lnc-ROR have confirmed this pathway showing that its inhibition is shown by suppression of invasiveness, migration, reduction in tumor size and reversion of drug resistance in BC cells [218,221,222]. However, its role in BCSCs and metastasis is unclear.

In addition to these, various lncRNAs such as LUCAT1, lncRNA-Hh, FGF13-AS1, lncRNA ES1 NEAT1 have been reported to be commonly involved in up-regulation of signaling pathways and modulation of stem cell factors (Wnt/β-catenin, Hedgehog, myc, SOX2, OCT4, KLF4, and NANOG). Their role in the promotion of stemness in BC cells and subsequent tumor progression, invasion and metastasis is critical for tumor maintenance and therapeutics [223,224,225,226,227].

The detailed role of lncRNAs in BC is described in Table 2.

## 6. Exosomal miRNAs: A Future Tool for Prognosis, Drug Discovery and As Therapeutic Targets

The significant presence of miRNAs was detected in biological fluids. miRNAs isolated from these sources are highly stable and non-degradable in extreme physiological conditions. It was reported that cells in culture transport intracellular miRNAs into the extracellular environment by exosomes [240]. Several studies revealed that these exosomal miRNA are implicated in cancer research, as tumor cells secrete different microRNAs capable of initiating cross-talk with the adjacent tumor microenvironment and educate them for adapting tumor favoring conditions for cancer progression [129,241,242,243,244,245]. Many exosomal miRNA were intensively studied for their ability to promote tumor progression by indicating drug resistance (miR-9,mir 221/222,miR 1246),metabolic reprogramming in CAF cells(miR105), intimating angiogenesis in endothelial cells(miR105, miR210), tumorigenesis in epithelial cells (miR10b, miR10a, miR21), osteogenesis in MSCs(mir940) [246,247,248,249]. Moreover, these exosomal miRNA can be circulated and used as potential diagnostic and prognostic markers in breast cancer [246,250]. For example, plasma and serum samples of breast cancer patients show microRNAs such as miR-106a-3p, 106a-5p, 20b-5p, and 92a-2-5p (plasma miRNAs); miR-106a-5p, 19b-3p, and 92a-3p (serum miRNAs) can be used as potential biomarkers in BC patients [251]. Some exosomal miRNA can be used as promising diagnostic markers, for example, high level of mir373 is associated with aggressive cancers, and a lower level of miR130-3p is associated with the advanced stage of cancer [252,253]. On the other hand, anticancer drugs derived from either natural or synthetic sources are reported to be dependent on miRNAs as exosomal cargoes to exert its anticancer activity [246]. For example, reduction in the growth of BC cells was associated with inhibition of secretion of exosomes containing miR-130a and miR-125 by D-rhamnose β-hederin, an oleanane type triterpenoid saponin [254]. Epigallocatechin gallate, one of the constituents present in green tea, induced its anti-cancer activity by up-regulating miR-16 in 4T1BC cells. [255] Chemosusceptibility was found to be elevated by β elemene by affecting the expression of miR-34a, miR-452, PTEN [256]. Shikonin a well-known natural compound exhibited antiproliferative effect by attenuating tumor-derived exosomal miR-128[257]. Docosahexanoic acid administration altered BC cells exosome secretion and microRNA content thereby inhibiting angiogenesis process [258]. 

Substantial evidence shows that exosomes act as a carrier and they could be manipulated to deliver tumor suppressor miRNA to exhibit their therapeutic potential [246]. Published studies have showcased that mesenchymal derived extravesicular vesicles can be successfully modified as a carrier for antitumor agents, to treat different forms of tumors [259]. The engineering of tumor-derived exosomes by electroporation method can help in overexpressing miR-155, -142, and let-7i, to mature dendritic cells and also to trigger the immunity process, to load siRNAs or miRNAs by sonication and also to knockdown oncogene such as HER2 [246]. Transfection of mesenchymal stem cells with anti-miR-222/223 transformed mesenchymal cells to dormant cancer cells and prolonged survival rate [260]. Gold-nanoparticle-facilitated RAB27A silencing in BC cells results in decreased exosomes secretion with no effect on cell viability. Exosomes were also reported to prevent tumor development both in vivo and in vitro [261,262]. Although some progress has been made to identify the potential of exosomal miRs in cancer research, it remains inconclusive as there is no standard technique reliable to isolate exosomes. The biomarker and drug therapy discoveries demand more detailed research in the field of exosomal micro RNA identification and classification.

In addition to the above techniques, using nanoparticles has also shown to increase stability and improved the delivery capability of miRNA. BC cell migration and invasion were inhibited by poly lysine-anti-miR10b complex [263]. Similarly, reduction in tumor growth capacity was observed when antisense miR-21 and antisense miR-10b were complexed with PLGA-b-PEG nanoparticle [264]. Encapsulation of miR34a with doxorubicin into hyaluronic acid chitosan successfully inhibited the migration of BC cells via the Notch-1 signaling pathway [265]. Designing various forms of nanoparticles such as gold, nano complex, and poly sorbitol-mediated transporter to carry the various form of miRNA not only improved delivery but also targeted and controlled cell proliferation of BC cells [165,266,267]. Cell cycle targeting miRNAs, miR-193a-3p and miR-214-5p encapsulated as nanoparticle showed high therapeutic potential against TNBC in vivo [268]. 

In light of the clinical impact, several miRNA-based therapies are under development whereas several of them are under pre-clinical and clinical stages. miRNA for treatment of pathologic fibrosis and blood cancer, non-small cell lung cancer and hepatocellular carcinoma is in the clinical stage however, not many lead molecules have been able to find their place either in pre-clinical or clinical trials for BC therapy. Looking at the potential of ncRNA targeting, we can assume that in the near future, the use of miRNA or lncRNA as mimics or inhibitor will be a suitable choice either alone or as an adjustment with existing therapeutic agents for regulating different aspects of human cancer [165].

In addition to the above therapies, the use of hormone therapy also known as endocrine therapy is considered to be a viable approach in point with detectable ER expression. The standard approach for treatment includes the use of tamoxifen for 5–10 years in pre-menopausal and a combination of tamoxifen with aromatase inhibition for post-menopausal women. Continuous use of tamoxifen is associated with the development of resistance; a newer viable strategy to overcome this issue is still underway [165].

The role of ncRNA in regulations of gene expression and BC implies it to be a potential target for treatment. However, data on ncRNA is still at its infancy stage with limited knowledge of its biological functions. Therefore, extensive research is required to understand its role as a prognostic, diagnostic or therapeutic target. 

## 7. Conclusions

Our review article has provided reports on extensive investigations and studies on the biological and functional role of miRNA and lncRNA in BC and CSCs providing an insight into their significance in cancer proliferation, pathological manifestations, progression invasion, and metastasis as biomarkers and as a potential therapeutic target. However, there are various considerations and challenges that need to be addressed. Firstly, in vivo studies, investigating the role of miRNAs in transgenic and knockout models are required to further ascertain their role in therapeutic targeting for the management of BC. Secondly, targeting breast cancer stem cells is a challenge in itself as accurate identification of reliable CSC markers as well as inherent heterogeneity of these cells hinders the targeting of signaling pathways by ncRNAs. Furthermore, knowledge of the types of lncRNA and their pathways in BC is still limited and extensive research to decipher its role as a biomarker/therapeutic targeting is needed. Therefore, large scale studies focusing on translational aspects of ncRNAs are required in order to fully understand and use its potential in BC treatment. 

## Figures and Tables

**Figure 1 cancers-12-00351-f001:**
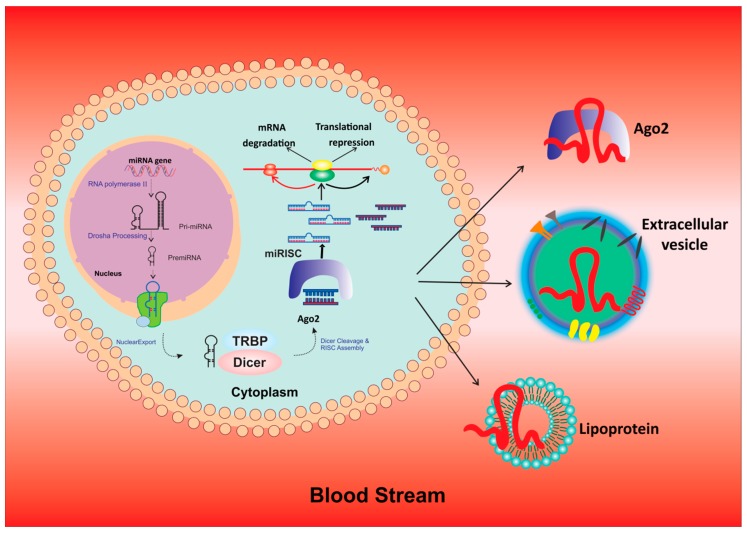
Process of biogenesis of miRNAs in the nucleus, its transfer into cytoplasm and functions.

**Figure 2 cancers-12-00351-f002:**
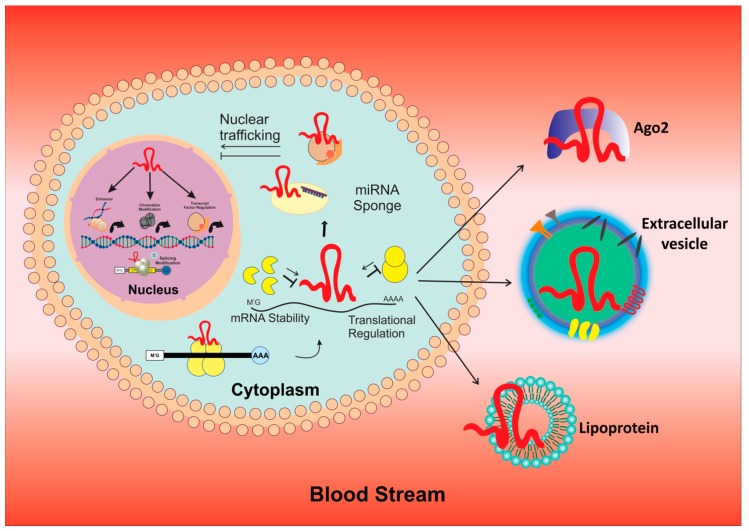
Illustrates the mechanism involved in process of biogenesis and function of lncRNA.

**Figure 3 cancers-12-00351-f003:**
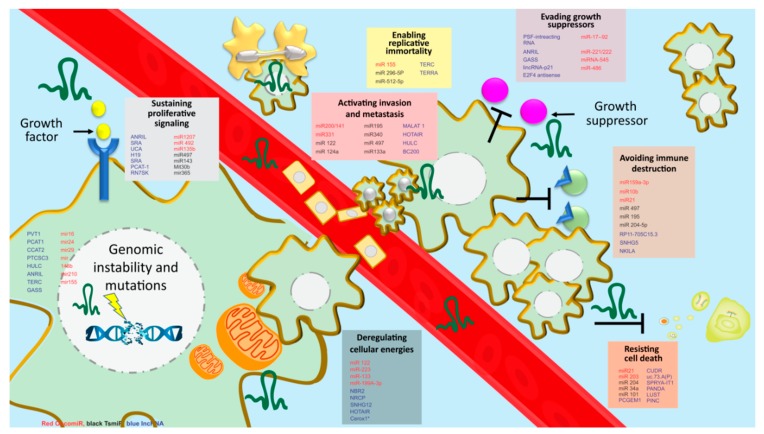
MicroRNA and LncRNA involved in breast cancer stemness therapy resistance and tumorigenesis. There are eight hallmarks implicated in cancer including sustaining proliferative signaling, enabling replicative immortality, evading growth suppressors, activating invasion and metastasis avoiding immune destruction, resisting cell death, deregulating cellular energetics and genomic instability and mutations. Expression of several microRNAs and lncRNA is associated with inducing oncogenic or tumor-suppressive properties via using the hallmarks of cancers.

**Table 1 cancers-12-00351-t001:** Role of miRNAs acting as tumor suppressor/oncomir in BC with their targeted pathways.

miRNA	Type	Expression Level	Targets	Pathways	Reference
miR-31	TsmiR	↑/↓	ITGA5, RDX, RHOA	Metastasis	[112,113]
miR-145	TsmiR	↓	MUC1, ERA, RTKN	Proliferation, Apoptosis, Invasion	[114,115,116]
miR-155	TsmiR	↑	FOXO3A, RHOA, SOCS1	STAT3, Proliferation, TGFβ Signaling	[117,118,119]
miR-21	OncomiR	↑	BCL2, PTEN, MMP3, TPM1, MASPIN, PDCD4, RHOB	EMT, Apoptosis, Invasion, Migration, Inflammatory Signals	[120,121,122,123,124]
miR-125b	TsmiR	↑/↓	BAK1, ERA, HER2, CRAF, RTKN, MUC1	Migration, Proliferation, Apoptosis	[125,126,127]
miR-10b	OncomiR	↑/↓	HDAC4, TIAM, HOXD10, EMT	EMT, Metastasis, Invasion	[128,129,130]
miR-205	TsmiR	↓	HER3, VEGFA, EMT	Proliferation, Invasion	[131,132,133]
miR-210	OncomiR	↑	MNT, RAD52	Hypoxia	[134,135]
miR-196A	OncomiR	↑	ANXA1	Proliferation, Apoptosis,	[136]
miR-944	OncomiR	↑	BNIP3	Cell Proliferation, Migration, Invasion	[137]
miR-222	OncomiR	↑	PTEN	PTEN, Akt/FOXP1	[138]
miR-3646	OncomiR	↑	GSK-3β	β Catenin	[139]
miR-34A	OncomiR	↑	BCL2, CCND1	Apoptosis	[140]
miR-141	OncomiR	↑	EIF4E	Apoptosis	[141]
miR-520h	OncomiR	↑	DAPK2	PI3K/Akt	[142]
miR-34	TsmiR	↓	BCL2, NOTCH	Apoptosis, NOTCH	[143]
miR-146	TsmiR	↓	NFkB	Inflammatory Signals	[144]
miR-7	TsmiR	↓	EGFR	EGFR	[145]
miR-22	TsmiR	↓	HER3, CDK6, ERα, CDC25C, SP1	Estrogen Signaling	[146]
miR-221	TsmiR	↑	P27, P57	Wnt/β-catenin	[147]
miR-191	OncomiR	↑	SATB1, CDK6, BDNF	Estrogen Signaling	[148]
miR-196A	OncomiR	↑	ANXA1	Apoptosis	[136]
miR-335	TsmiR	↑	SOX4, TNC, PTPRN2, MERTK	Metastasis	[149]
miR-20	OncomiR	↑	E2F	Proliferation	[150]
miR-9	TsmiR	↑	LIFR, E-CADHERIN	EMT, Hippo-YAP	[151,152]
miR-126	TsmiR	↓	VEGFA and PIK3R2	VEGF/PI3K/AKT	[153]
miR-98	TsmiR	↑	ALK4 and MMP11	Angiogenesis, Invasion	[154]
miR-148a/152	TsmiR	↓	DNMT1, IGF-IR and IRS1	IGF-IR/PKM2	[155]
miR-519c	TsmiR	↓	HIF-1α	Hypoxia	[156]
miR-10b	OncomiR	↑	HOXD10	Hox pathway	[157]
miR-140-5p	TsmiR	↓	VEGFA	Metastasis, Angiogenesis	[158]
miR-494	TsmiR	↑	PTEN	Akt, NF-kB, mTOR	[159]
miR-206	TsmiR	↓	VEGF, MAPK3, and SOX9	Invasion, Angiogenesis	[160]
miR-19a	OncomiR	↑	PTEN	Oncogenic PTEN Cell proliferation, Th1 immune response	[161]
miR-17-92	TsmiR	↓	HIF-1α	Hypoxia, Angiogenesis.	[162]
miR-467	OncomiR	↑	TSP-1	Angiogenesis	[163,164]
miR-18	OncomiR	↑	SMAD7	EMT, Metastasis	[165]
miR-143	OncomiR	↑	FOSL2	EMT, Metastasis	[165]
miR-196B	OncomiR	↑	HOXD10	Hox pathway	[157]
miR-200	OncomiR	↑	ZEB1, ZEB2	EMT	[165]
miR-205	TsmiR	↓	YAP1	miR-205/YAP1, Angiogenesis, Metastasis	[166]
miR-892b	TsmiR	↑	TRAF2, TAK1, and TAB3	NF-kB	[167]
miR-210 RAD52	OncomiR	↑	RAD52	Invasion, Proliferation, Migration	[168]
mirR-155	OncomiR	↑	SOC6	STAT3 signaling	[169]
miR-451	OncomiR	↑	Bcl-2	Apoptosis	[170]
miR-100	OncomiR	↑	mTOR	Cell proliferation, Survival	[171]
miR-139-5p	OncomiR	↑	Notch1	Cell growth, Apoptosis	[172]
miR-214	OncomiR	↑	UCP2	Autophagy	[173]
miR-16	OncomiR	↑	CCNJ, FUBP1	PI3K/Akt	[174]
miR-199a-3p	TsmiR	↑	TFAM	Mitochondrial Biogenesis	[175]
miR-302b	TsmiR	↑	E2F1	E2f1-ATM axis	[176]
miR-218	TsmiR	↑	BRCA1	DNA repair, Cell proliferation, Invasion	[177]
miR-638	TsmiR	↑	BRCA1	DNA repair, Cell proliferation, Invasion	[178]
miR-29A	OncomiR	↑	PTEN	Apoptosis	[179]
miR-129-3p	OncomiR	↑	CP110	Apoptosis \, Cell Cycle, Cell Proliferation	[180]
miR-19	OncomiR	↓	Tissue factor	Angiogenesis, Metastasis	[181]

**Table 2 cancers-12-00351-t002:** Role of lncRNAs acting as either tumor suppressor/oncogene in BC with their targeted pathways.

lncRNA	Type	Expression Level	Targets	Pathways	Reference PMID
MEG 3	Tumor suppressor	↓	p53	p53	[228]
HOTAIR	Oncogene	↑	BRCA1, PTEN	PI3K/AKT-BAD pathway, HOXD10	[229]
ACNR	Tumor suppressor	↓	TGF-β	Metastasis, Invasion	[230]
PTENP1	Tumor suppressor	↓	PTEN	Apoptosis	[228]
NKILA	Oncogene	↓	NF-kB	EMT	[231]
EPIC 1	Oncogene	↑	Myc	Cell Cycle	[232]
PLNCRNA-1	Oncogene	↓	TGF-β	Apoptosis, Metastasis, Invasion	[228]
H19	Oncogene	↑	C-myc	AKT, BIK	[233,234]
MALAT-1	Oncogene	↑/↓	AKT, p53	APOPTOSIS	[235]
LINK-A	Oncogene	↑	HIF-1α	Hypoxia Pathway	[228]
CCAT2	Oncogene	↑	ERK	MAPK	[236]
PVT-1	Oncogene	↑	KLF-5,β-Catenin	WNT/β-Catenin	[228]
UCA1	Oncogene	↑	mTOR,β-Catenin	mTOR, WNT/ β-Catenin	[237,238]
GAS5	Tumor suppressor	↓	PTEN	Apoptosis	[239]
BCAR4	Oncogene	↑	SNIP1, PNUTS	Hedgehog /GLI 2 Signaling Transduction	[228]
NEAT	Oncogene	↑	ZEB1, RAS	RAS, MAPK, RSF1	[227]

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
