# Peer review of "Non-Coding RNAs as Regulators and Markers for Targeting of Breast Cancer and Cancer Stem Cells"

_cancers, 2020, doi:10.3390/cancers12020351_

Round 1

Reviewer 1 Report

This manuscript submitted by Prabhu et al. aims to provide a review on roles of non-coding RNAs in breast cancer. This topic is quite interesting. However, the manuscript is poorly written. It is not just poor grammar, syntax and style. The manuscript shows lack of knowledge in RNA biology in general and lncRNA biogenesis in particular. Even general biology statements in this manuscript are questionable. Just one example, Line 155 claims that “… breast tissue is a highly heterogeneous organ…” Tissue is not an organ! Even undergrads know that. This is a review paper. It should not provide any wrong or misleading information.

It is clear that the authors have no expertise in the field they are writing about. I do believe the team of 13 authors spent some time reading and analyzing the literature, but that does not immediately qualify them for writing a decent review article on a new subject.

Author Response

Reviewer 1

This manuscript submitted by Prabhu et al. aims to provide a review on roles of non-coding RNAs in breast cancer. This topic is quite interesting. However, the manuscript is poorly written. It is not just poor grammar, syntax and style. The manuscript shows lack of knowledge in RNA biology in general and lncRNA biogenesis in particular. Even general biology statements in this manuscript are questionable. Just one example, Line 155 claims that “… breast tissue is a highly heterogeneous organ…” Tissue is not an organ! Even undergrads know that. This is a review paper. It should not provide any wrong or misleading information.

 It is clear that the authors have no expertise in the field they are writing about. I do believe the team of 13 authors spent some time reading and analyzing the literature, but that does not immediately qualify them for writing a decent review article on a new subject.

Authors Response (AR): We appreciate for the critical feedback and as per reviewer suggestion we have now edited our revised manuscript extensively to improve grammar, syntax and style. We have also worked upon to improve general biology statements so as to avoid any misleading information.

Our team comprised of eminent scientist with expertise in the field of microRNA, cancer biology, immunology, breast cancer surgery with several years of experiences and having publications in renowned journals. Below we have mentioned some of articles published by authors in reputed journal of international standard.

Thasni Karedath (Publications: Out of 20)

1.Ahmed I, Karedath T, Al-Dasim FM, Malek JA. Identification of human genetic variants controlling circular RNA expression. RNA. 2019;25(12):1765-78.

2.Al Ameri W, Ahmed I, Al-Dasim FM, Ali Mohamoud Y, Al-Azwani IK, Malek JA, et al. Cell Type-Specific TGF-beta Mediated EMT in 3D and 2D Models and Its Reversal by TGF-beta Receptor Kinase Inhibitor in Ovarian Cancer Cell Lines. Int J Mol Sci. 2019;20(14).

3.Karedath T, Ahmed I, Al Ameri W, Al-Dasim FM, Andrews SS, Samuel S, et al. Silencing of ANKRD12 circRNA induces molecular and functional changes associated with invasive phenotypes. BMC Cancer. 2019;19(1):565.

4.Ahmed I, Karedath T, Andrews SS, Al-Azwani IK, Mohamoud YA, Querleu D, et al. Altered expression pattern of circular RNAs in primary and metastatic sites of epithelial ovarian carcinoma. Oncotarget. 2016;7(24):36366-81.

Kirti.S.Prabhu (Publications: Out of 20)

1.Halama A, Kulinski M, Dib SS, Zaghlool SB, Siveen KS, Iskandarani A, et al. Accelerated lipid catabolism and autophagy are cancer survival mechanisms under inhibited glutaminolysis. Cancer Lett. 2018;430:133-47.

2. Khan AQ, Kuttikrishnan S, Siveen KS, Prabhu KS, Shanmugakonar M, Al-Naemi HA, et al. RAS-mediated oncogenic signaling pathways in human malignancies. Semin Cancer Biol. 2019;54:1-13.

3.Siveen KS, Prabhu KS, Achkar IW, Kuttikrishnan S, Shyam S, Khan AQ, et al. Role of Non Receptor Tyrosine Kinases in Hematological Malignances and its Targeting by Natural Products. Mol Cancer. 2018;17(1):31.

4. Prabhu KS, Siveen KS, Kuttikrishnan S, Jochebeth A, Ali TA, Elareer NR, et al. Greensporone A, a Fungal Secondary Metabolite Suppressed Constitutively Activated AKT via ROS Generation and Induced Apoptosis in Leukemic Cell Lines. Biomolecules. 2019;9(4).

Afsheen Raza (Publications: Out of 26)

1.Maacha S, Bhat AA, Jimenez L, Raza A, Haris M, Uddin S, et al. Extracellular vesicles-mediated intercellular communication: roles in the tumor microenvironment and anti-cancer drug resistance. Mol Cancer. 2019;18(1):55. 2. Merhi M, Raza A, Inchakalody V, Zar AR, Uddin S, Dermime S. Immunotherapeutic strategies in patients with advanced head and neck squamous cell carcinoma. Ann Transl Med. 2019;7(Suppl 1):S22.

3.Siveen KS, Raza A, Ahmed EI, Khan AQ, Prabhu KS, Kuttikrishnan S, et al. The Role of Extracellular Vesicles as Modulators of the Tumor Microenvironment, Metastasis and Drug Resistance in Colorectal Cancer. Cancers (Basel). 2019;11(6).

4.Fernandes Q, Merhi M, Raza A, Inchakalody VP, Abdelouahab N, Zar Gul AR, et al. Role of Epstein-Barr Virus in the Pathogenesis of Head and Neck Cancers and Its Potential as an Immunotherapeutic Target. Front Oncol. 2018;8:257.

Said Dermime (Publications: Out of 62)

1.Mohsen MO, Vogel M, Riether C, Muller J, Salatino S, Ternette N, et al. Targeting Mutated Plus Germline Epitopes Confers Pre-clinical Efficacy of an Instantly Formulated Cancer Nano-Vaccine. Front Immunol. 2019;10:1015.

2.Xu Y, Wan B, Chen X, Zhan P, Zhao Y, Zhang T, et al. The association of PD-L1 expression with the efficacy of anti-PD-1/PD-L1 immunotherapy and survival of non-small cell lung cancer patients: a meta-analysis of randomized controlled trials. Transl Lung Cancer Res. 2019;8(4):413-28.

3.Khan, A. Q., K. S. Siveen, K. S. Prabhu, S. Kuttikrishnan, S. Akhtar, A. Shaar, et al. Curcumin-Mediated Degradation of S-Phase Kinase Protein 2 Induces Cytotoxic Effects in Human Papillomavirus-Positive and Negative Squamous Carcinoma Cells. Front Oncol (2018); 8: 399.

Martin Steinhoff (Publications: Out of 568)

1.Brodsky AS, Fischer A, Miller DH, Vang S, MacLaughlan S, Wu HT, et al. Expression profiling of primary and metastatic ovarian tumors reveals differences indicative of aggressive disease. PLoS One. 2014;9(4):e94476. 2.Matei D, Filiaci V, Randall ME, Mutch D, Steinhoff MM, DiSilvestro PA, et al. Adjuvant Chemotherapy plus Radiation for Locally Advanced Endometrial Cancer. N Engl J Med. 2019;380(24):2317-26.

3.Steinhoff M, Schmelz M, Szabo IL, Oaklander AL. Clinical presentation, management, and pathophysiology of neuropathic itch. Lancet Neurol. 2018;17(8):709-20.

Shahab Uddin(Publications: Out of 200)

1 Khan AQ, Ahmed EI, Elareer NR, Junejo K, Steinhoff M, Uddin S. Role of miRNA-Regulated Cancer Stem Cells in the Pathogenesis of Human Malignancies. Cells. 2019;8(8).

2.  Bhat AA, Uppada S, Achkar IW, Hashem S, Yadav SK, Shanmugakonar M, et al. Tight Junction Proteins and Signaling Pathways in Cancer and Inflammation: A Functional Crosstalk. Front Physiol. 2018;9:1942.

3.  Khan AQ, Siveen KS, Prabhu KS, Kuttikrishnan S, Akhtar S, Shaar A, et al. Curcumin-Mediated Degradation of S-Phase Kinase Protein 2 Induces Cytotoxic Effects in Human Papillomavirus-Positive and Negative Squamous Carcinoma Cells. Front Oncol. 2018;8:399.

4. Siveen, K. S., S. Uddin, and R. M. Mohammad. "Targeting Acute Myeloid Leukemia Stem Cell Signaling by Natural Products." Mol Cancer.2017; 16:13.

We hope the reviewer will appreciate  the accomplishment of our team and collaborators of this review article.

Reviewer 2 Report

This manuscript reviews biogenesis and functional roles of micro RNA and long non-coding RNAs in breast cancer and breast cancer stem cells. This review well covers the topic and well written.

Line 221, Section 5 “target various signaling pathways such as PI3Kinase, Wnt/βcatenin, STAT3, HIF 1α etc. [88-94]. Detailed role of these ncRNAs is depicted in Figure 3. However, there is no such signaling pathways indicated in Fig. 3. Also, no such signaling pathways were described in section 6 (Role in microRNA).

In section 8 (Non-coding RNA as a therapeutic target in BC), the references 233 – 246 don’t directly relate to therapy. In these papers, CRISPR, siRNA etc. are used for functional assays.

Author Response

This manuscript reviews biogenesis and functional roles of micro RNA and long non-coding RNAs in breast cancer and breast cancer stem cells. This review well covers the topic and well written.

AR: We would like to thank reviewer for mentioning that “This review well covers the topic and well written”.

Line 221, Section 5 “target various signaling pathways such as PI3Kinase, Wnt/βcatenin, STAT3, HIF 1α etc. [88-94]Detailed role of these ncRNAs is depicted in Figure 3. However, there is no such signaling pathways indicated in Fig. 3. Also, no such signaling pathways were described in section 6 (Role in microRNA).

AR: We agree with reviewer and have now included all the suggestions in the revised manuscript. Now we have described the role of various signaling pathways under Role of microRNAs and lncRNA in BCSCs(section 5). We also agree that no such signaling pathways are indicated in Fig 3. Figure 3 only enlist various oncomiR and TsmiR that play a significant role in hallmarks of cancers. However, revised text inserted in Section 5 as well as Table 1, allows the reader to understand the targets and signaling pathways associated with this figure.

In section 8 (Non-coding RNA as a therapeutic target in BC), the references 233 – 246 don’t directly relate to therapy. In these papers, CRISPR, siRNA etc. are used for functional assays.

AR: We agree with reviewer and have now included all the suggestions in the revised manuscript. We have deleted the references that don’t directly relate to Section 8.

Reviewer 3 Report

This review titled " Non-coding RNAs as regulators and markers for targeting of breast cancer and cancer stem cells " is quite exaustive and interesting. This review manuscript by

Prabhu and collegues summarizes recent literature regarding the novel findings about the emerging role of lncRNAs and miRNAs in tumor progression, metastasis and stemness. In particular the authors addressed the aspect of non-coding RNAs as biomarkers, eventually targeted, in breast cancer (BC).

The work could lack of originality because in the scientific literature many reviews about non-coding RNAs in BC that function as novel biomarkers to diagnose or predict prognosis are present. However, this review has the merit of being very well written, the tables are well detailed.

To render this manuscript more original and suitable for the publication, I would suggest to state the function of diverse miRNAs in exosomes mediated cell-cell communication and the potency of some specific enriched miRNAs as molecular markers in clinical trials. The authors may also describe the mechanism of anti-cancer compounds through exosomes and the exploration of artificially engineered techniques that lead miRNA-inhibitors into exosomes for therapeutic use.

Author Response

Reviewer 3:

This review titled “Non-coding RNAs as regulators and markers for targeting of breast cancer and cancer stem cells “is quite exhaustive and interesting. This review manuscript by Prabhu and collegues summarizes recent literature regarding the novel findings about the emerging role of lncRNAs and miRNAs in tumor progression, metastasis and stemness. In particular the authors addressed the aspect of non-coding RNAs as biomarkers, eventually targeted, in breast cancer (BC).The work could lack of originality because in the scientific literature many reviews about non-coding RNAs in BC that function as novel biomarkers to diagnose or predict prognosis are present. However, this review has the merit of being very well written, the tables are well detailed.

AR: We would like to thank reviewer for their appreciative comments and mentioning that the review is exhaustive, interesting and has the merit of being very well written with well detailed tables.

To render this manuscript more original and suitable for the publication, I would suggest to state the function of diverse miRNAs in exosomes mediated cell-cell communication and the potency of some specific enriched miRNAs as molecular markers in clinical trials. The authors may also describe the mechanism of anti-cancer compounds through exosomes and the exploration of artificially engineered techniques that lead miRNA-inhibitors into exosomes for therapeutic use.

AR: We agree with reviewer and have now included all the suggestions under Section 8 titledExosomal miRNA’s a future tool for BC prognosis, drug discovery and as therapeutic targets”.

Based on the extensive revisions made as per reviewers suggestions, we believe that the manuscript has now improved immensely and is suitable for publication in its revised format. We hope that the journal will accept it for publication under schematic theme "Non-Coding RNAs as Emerging Regulators of Signaling Pathways and Novel Therapeutic Targets in Human Cancers.

Round 2

Reviewer 1 Report

The revised version does not show much improvement. It is still poorly written. The main purpose of this type of publications is to give an overview and/or expert opinion on major findings in a particular field of research for people who are not familiar with the topic, such as students and scientists exploring a new niche. The question is if this manuscript provides useful and reliable information. And my answer is NO. Personally, I would not recommend such a review article for reading.

Author Response

Reviewer 1 comment

The revised version does not show much improvement. It is still poorly written. The main purpose of this type of publications is to give an overview and/or expert opinion on major findings in a particular field of research for people who are not familiar with the topic, such as students and scientists exploring a new niche. The question is if this manuscript provides useful and reliable information. And my answer is NO. Personally, I would not recommend such a review article for reading.

Authors Response to reviewer 1 comments:

We would like to thank the reviewer for their comments. As per the first round comments, we had revised the manuscript and added new information such as modulation of signaling pathways by miRNAs influencing cancer hallmarks and an overview on exosomal miRNA’s as a future tool for prognosis, drug discovery and therapeutic targets. Furthermore, we also addressed issues related with syntax and grammatical errors.

         We respectfully disagree with the reviewer comments on the manuscript not providing useful or reliable information to students and scientists. As per our understanding, we have provided useful overview on major findings in this field with the help of recent publications and literature.

  We believe that the manuscript is much improved and is suitable for publication

Reviewer 3 Report

The manuscript has been significantly improved. The new paragraph increased the novelty of the work. In this version it is acceptable for publication.

Author Response

We thank reviewer for mentioning "manuscript has been significantly improved" and version it is acceptable for publication.